# Individual Experiences with Being Pushed to Limits and Variables That Influence the Strength to Which These Are Felt: A Cross-Sectional Survey Study

Eisuke Nakazawa [1], Katsumi Mori [1] and Akira Akabayashi [1,2,*]

1 Department of Biomedical Ethics, Faculty of Medicine, University of Tokyo, 7-3-1 Hongo, Bunkyo-ku, Tokyo 113-0033, Japan
2 Division of Medical Ethics, School of Medicine, New York University, 227 East 30th Street, New York, NY 10016, USA
* Correspondence: akirasan-tky@umin.ac.jp or akira.akabayashi@gmail.com; Fax: +81-35841-3319

**Abstract:** In a 2021 survey, we found that "limit or suppression experiences" were related to a willingness to use enhancement technologies. However, the concept of "limit or suppression experiences" is vague and difficult to interpret in relation to neuroethics/enhancement. Thus, we aimed to better understand "limit or suppression experiences" and establish a robust philosophical concept of the topic. To do so, we exploratively investigated the concept to determine individual experiences with the presence or absence of sensing limits, investigate different ways in which limits can be sensed (factors of the sense of limits: "FSLs"), and identify factors that correlate with the strength of FSLs. Data from an Internet survey investigating respondents' experiences with limits (1258 respondents) were analyzed using exploratory factor analysis and a linear regression model. Five variables were extracted as the FSLs. The highest regression coefficients were found between physical FSL and sports activities and between cognitive FSL and academics. The lowest regression coefficients were found between relational FSL and academics, sports activities, and arts and cultural activities. The results facilitate a detailed discussion of the motivations of enhancement users, and the extraction of the suppression experience opens new enhancement directions. Further normative and empirical studies are required.

**Keywords:** limit or suppress experience; sense of limit; neuroethics; enhancement; Internet survey; factor analysis; multiple logistic regression analysis; medical care settings

## 1. Introduction

Human enhancement refers to attempts to overcome the current limits of the human body using techniques designed or applied to restore or improve human capabilities [1,2]. Human enhancement is often categorized into physical, cognitive, and moral (emotional) enhancement [3,4]. Users' enhancement motivation is significant for the direction of enhancement technology's development. Prior studies have reported on enhancement motivation for cognitive and mood enhancement using drugs, arguing for the prevention of drug misuse [5–7]. However, research on enhancement motivation is still limited; thus, a comprehensive discussion is needed to extend the research to include physical enhancement and interventions using neuroscience techniques that go beyond pharmacological interventions. In fact, minimally invasive brain intervention technologies, including neurofeedback, are being rapidly developed, and their practical use may soon become a reality. Therefore, we previously conducted an online survey in Japan [8]. Approximately 20% of respondents in that study reported that they were willing to use enhancement technologies, whereas 80% were not. Using a generalized linear mixed-effects model, we examined the association between the type of intervention and the respondents' willingness to use such technologies. Factors related to the willingness to use these technologies included the intervention's degree of invasiveness, as well as respondents' gender, educational attainment,

and "limit or suppression experiences." In that study, the term "limit" had been arbitrarily defined as the individual's "inability to reach a better condition, despite all efforts toward achieving a given goal or purpose"; the term "suppression" was also arbitrarily defined as the individual "realizing what their limits are before reaching them, stopping further efforts, and compromising."

The concept of a "limit" in the definition of human enhancement can be applied to the discussion of enhancement motivation. A limit, in the broadest sense, is a point at which human effort reaches an insurmountable barrier. Individuals experience various limits in their lives. People living in modern society constantly experience physical, emotional, and social stresses [9,10]. When studying for entrance exams, students experience severe emotional stress, and athletes endure rigorous practices that generate physical stress. Life events and stages, such as marriage, childbirth, and childrearing, are no exception, generating their own assortment of stress [11]. However, human beings are not omnipotent; thus, when they are striving to achieve a goal, they will eventually reach their limits, at which point they will feel it is no longer possible to continue. Individuals respond differently when approaching their limits. Some may push past their limits and give up, and some of them may even become ill [12–18].

Historically, Karl Jaspers made the concept of limit an explicit theme and examined it philosophically and psychologically. Jaspers described that "the ultimate situations—eath, chance, guilt, and the uncertainty of the world—confront me with the reality of failure [ . . . . . . ]. And yet the Stoics' striving is toward true philosophy. Their thought, because its source is in ultimate situations, expresses the basic drive find a revelation of true being in human failure [19]." Further, Jaspers identified the concept of limit as the fundamental energy that motivates the essential life of human beings. Jaspers' concept of limit has been reevaluated in recent years, and its psychotherapeutic implication and applicability have been indicated in some research [20].

The concept of limit (or limitation) was also developed in the field of sports science as a content-correlated but possibly separate source. According to the classic studies of Ikai et al. [21,22], the physiological limit of muscle strength is the upper limit of muscle strength determined by structural factors, and the psychological limit is the upper limit of muscle strength determined by functional factors. Ikai et al.'s studies opened up a research area on the subjective aspect of muscle strengthening and fitness [23–27], which has resulted in research on fatigue [28–30], physical exercise, and encouragement [28,31,32]. In recent years, studies on intervening in the psychological aspects of physical exercise to improve performance, including neurofeedback interventions, have also been conducted [22–35], and Ikai et al.'s concept of limit [21,22] is becoming more relevant to contemporary enhancement technologies.

We thus assumed that the concept of limit or suppression experiences could be a useful cross-cutting concept in the discussion of motivation as a foundational theory of enhancement and attempted to clarify it. Given this context, as the first step in understanding and establishing the concept of "limit or suppression experiences" related to neuroethics/enhancement, the present study exploratively investigated the concept of limits to clarify what "limit or suppression experiences" actually means. The survey aimed to (1) ask about individual experiences with the presence or absence of sensing limits, (2) investigate the different ways in which limits can be sensed (factors of the sense of limits: "FSLs"), and (3) identify factors that correlate with the strength of FSLs. When discussing the policy of use of the enhancement technique, it is crucial to understand and establish the meaning of limit or suppression experiences. Therefore, the aim of this study was preliminary without any hypothesis, but we have attempted to illustrate the entity of limit or suppression experiences as much as possible.

## 2. Respondents and Methods

### 2.1. Respondents and Survey

In March 2021, a questionnaire was e-mailed to all individuals aged 20 years and older across Japan who were registered as questionnaire response members (monitors) of

a major Internet survey company in Japan. Responses were accepted until the required number of responses for each sex and age group were received. Overall, responses from 1258 respondents were included. Completion of the questionnaire was considered as consent to participate in this study.

Questionnaire Content

The term "limit" was defined as "when striving to achieve a goal or objective, reaching the point at which no further improvements were possible, even when exerting maximum effort to achieve a goal or objective" (hereafter termed limit experience, abbreviated as LE). "Suppression when a limit is anticipated" was defined as "before arriving at the limit, to foresee one's limit, stop trying any further, and compromise" (hereafter termed suppression experience, abbreviated as SE). Respondents were asked whether they had experienced LE or SE. Those who responded "yes" were asked about the type of activity that led to the experience, such as work/housework/childrearing, academics, sports activities (e.g., sports, dancing, and hiking), and arts and cultural activities (e.g., drawing, music, and theater).

Respondents who had experienced LE or SE were asked about the limits that left the biggest impression on them. With regard to each of the 29 items listed regarding skills (e.g., memory) and characteristics (e.g., kindness), they were asked to rate the degree to which they felt limited when unable to achieve the goal or objective (i.e., when the limit was experienced). Ratings were made on a 5-point scale comprising the following: "Did not require it for this experience," "Did not feel this at all," "Felt it a little bit," "Felt it a fair bit," and "Felt it very strongly." The 29 items were determined by the authors, who referenced previous studies and expert interviews [36–44]. Those who did not experience either LE or SE (hereafter, no experience: NE) were asked why they never had such experiences, offering five different reasons as options. All respondents were inquired regarding their demographic and socioeconomic backgrounds. Prior to the survey, a pre-test of the questionnaire was conducted with five people from the general population to ensure that all questions or expressions could be easily understood and that responses could be given in a straightforward manner.

*2.2. Analysis*

Data were collected on respondent characteristics, the presence or absence of LE or SE, the setting of the LE or SE (for those with LE or SE), and the reasons for the lack of such experiences (NE individuals only). To examine characteristic-based differences in the presence/absence of LE or SE, data were stratified by background characteristics, noting whether the respondent had LE or SE.

To examine FSLs, respondents who had an LE or SE were targeted and asked to specify the degree to which they felt limited in that particular experience using exploratory factor analysis of the 29 response items. Selection of either "Did not require it for this experience" or "Did not feel this at all" was considered to be indicative of the lack of a limit; accordingly, these were combined into one category. Using a polychoric correlation coefficient matrix [45] determined based on the answers from the resulting 4-point scale, FSLs were extracted using iterative principal factor analysis with varimax rotation.

To identify the variables associated with individual characteristics with regard to the extracted FSLs, LE, and SE, data were analyzed using a linear regression model with the factor score obtained from factor analysis as the objective variable, and the distinction between LE and SE, sex, age, highest academic degree, annual household income, and residential area (urban or non-urban) as the explanatory variables.

Statistical analyses were conducted using SAS version 9.4 (SAS Institute Inc., Cary, NC, USA). Statistical significance was set to be $p < 0.05$.

*2.3. Ethical Considerations*

The present study was approved by the appropriate research ethics committee (approval no. 10950).

**3. Results**

*3.1. Respondent Characteristics*

Sampling was conducted in a manner that would yield roughly the same number of responses from both sexes. Similarly, sampling was conducted so that each age group in 10-year increments, from 20 years and above, would have roughly the same number of respondents. In the analysis below, however, respondents were divided into three age groups: 20–39 years, 40–64 years, and 65 years and older (Table 1).

**Table 1.** Respondent characteristics and experiences with limits.

|  |  | *n* | (%) |
|---|---|---|---|
| Sex | Male | 618 | (49.1%) |
|  | Female | 640 | (50.9%) |
| Age | 20–39 years | 403 | (32.0%) |
|  | 40–64 years | 551 | (43.8%) |
|  | ≥65 years | 304 | (24.2%) |
| Highest academic degree | Junior high/high school | 414 | (32.9%) |
|  | Trade school, two-year college, or higher professional school | 276 | (21.9%) |
|  | University/graduate school | 556 | (44.2%) |
|  | Prefer not to respond | 12 | (1.0%) |
| Household annual income | <JPY 4 million | 432 | (34.3%) |
|  | JPY 4–8 million | 377 | (30.0%) |
|  | ≥JPY 8 million | 187 | (14.9%) |
|  | Not sure/prefer not to respond | 262 | (20.8%) |
| Residence | Non-urban setting | 802 | (63.8%) |
|  | Urban setting | 456 | (36.2%) |
| Experiences with limits | Have limit experience (LE) | 401 | (31.9%) |
|  | No limit experience (LE), but have suppression experience (SE) | 403 | (32.0%) |
|  | No experience with LE or SE (NE) | 454 | (36.1%) |
| Setting of limit experience [a] | Work/housework/childrearing | 279 | (34.7%) |
|  | Academics | 218 | (27.1%) |
|  | Sports activities | 175 | (21.8%) |
|  | Arts and cultural activities | 71 | (8.8%) |
|  | Other | 61 | (7.6%) |

[a] Those with either limit experience (LE) or suppression experience (SE): *n* = 804; other: *n* = 1258.

Respondents with LE comprised approximately 32% of the total respondents, and those with no LE but SE also comprised 32%, while those with NE comprised a slightly larger percentage (36.1%). The most common setting for both LE and SE was work/housework/childrearing (34.7%), followed by academics and sports activities.

*3.2. The Presence/Absence of LE and SE Stratified by Respondent Characteristics*

Differences in experiences were observed based on sex, as a higher percentage of female respondents indicated they "Have had neither LE nor SE" than their male counterparts; additionally, a lower percentage of female respondents answered, "Have had an LE" (Table 2). A significant age-dependent difference was also noted, with higher percentages of older groups comprising NE individuals and lower percentages of LE individuals. With regard to the highest academic degree obtained, those with higher degrees comprised a lower percentage of NE individuals and a higher percentage of LE individuals. When stratified by annual household income, a significant difference was observed, whereby a low percentage of NE individuals was identified among those with an annual income of ≥JPY 8 million. No significant differences were observed with regard to the residential area.



**Table 2.** Presence or absence of limit experience (LE) or suppression experience (SE) among respondents, stratified by basic characteristics.

| | | Have LE | | No LE, but Have SE | | Have Neither | | |
|---|---|---|---|---|---|---|---|---|
| | | *n* | (%) | *n* | (%) | *n* | (%) | *p*-Value [a] |
| Sex | Male | 210 | (34.0%) | 207 | (33.5%) | 201 | (32.5%) | 0.034 |
| | Female | 191 | (29.8%) | 196 | (30.6%) | 253 | (39.5%) | |
| Age | 20–39 years | 136 | (33.7%) | 147 | (36.5%) | 120 | (29.8%) | 0.001 |
| | 40–64 years | 189 | (34.3%) | 159 | (28.9%) | 203 | (36.8%) | |
| | ≥65 years | 76 | (25.0%) | 97 | (31.9%) | 131 | (43.1%) | |
| Highest academic degree | Junior high/high school | 109 | (26.3%) | 124 | (30.0%) | 181 | (43.7%) | <0.001 [b] |
| | Trade school, 2-year college, or higher professional school | 87 | (31.5%) | 84 | (30.4%) | 105 | (38.0%) | |
| | University/ graduate school | 203 | (36.5%) | 193 | (34.7%) | 160 | (28.8%) | |
| | Prefer not to answer | 2 | (16.7%) | 2 | (16.7%) | 8 | (66.7%) | |
| Household annual income | < JPY 4 million | 131 | (30.3%) | 146 | (33.8%) | 155 | (35.9%) | 0.018 [b] |
| | JPY 4–8 million | 123 | (32.6%) | 110 | (29.2%) | 144 | (38.2%) | |
| | ≥ JPY 8 million | 65 | (34.8%) | 75 | (40.1%) | 47 | (25.1%) | |
| | Not sure/Prefer not to answer | 82 | (31.3%) | 72 | (27.5%) | 108 | (41.2%) | |
| Residential area | Non-urban setting | 252 | (31.4%) | 256 | (31.9%) | 294 | (36.7%) | 0.839 |
| | Urban setting | 149 | (32.7%) | 147 | (32.2%) | 160 | (35.1%) | |

[a] Pearson's $\chi^2$ test; [b]: excludes those who responded "Not sure" and "Prefer not to respond".

### 3.3. Reasons for Not Having LE or SE

The responses provided by NE individuals when asked why they had no LE or SE are shown in Table 3 (*n* = 454). The most common response was "Only try to do what can be done at that given moment" (*n* = 185, 40.7%), followed by "Never had any goal or objective for which there was a desire to fulfill to the point of striving to the maximum limit" (*n* = 140, 30.8%). Other answers were: "With hard work, all goals and objectives could be achieved without feeling pushed to limits" (*n* = 76, 16.7%); "Always had to give up on goals and objectives because of external factors and societal situations" (*n* = 41, 9.0%); "Other" (*n* = 12, 2.6%).

**Table 3.** Reasons for not having LE or SE.

| | No. of Individuals | (%) |
|---|---|---|
| With hard work, all goals and objectives could be achieved without feeling pushed to limits | 76 | (16.7%) |
| Never had any goal or objective for which there was a desire to fulfill to the point of striving to the maximum limit | 140 | (30.8%) |
| Only try to do what can be done at that given moment | 185 | (40.7%) |
| Always had to give up on goals and objectives due to external factors and societal situations | 41 | (9.0%) |
| Other | 12 | (2.6%) |

Respondents with neither LE nor SE: *n* = 454.

### 3.4. Exploratory Factor Analysis

Table 4 shows the results of the exploratory factor analysis of LE and SE individuals (*n* = 804). The following five FSLs were extracted in descending order of factor loading (>0.45): relational FSL, physical FSL, cognitive FSL, psychological FSL, and technical skills FSL. Higher scores for these factors indicate that a stronger sense of limit was felt.

**Table 4.** Factor analysis of sense of limits.

| | Factor 1 | Factor 2 | Factor 3 | Factor 4 | Factor 5 |
|---|---|---|---|---|---|
| | Relational | Physical | Cognitive | Psychological | Technical skills |
| Contribution | 6.424 | 3.610 | 3.333 | 2.732 | 2.264 |
| Receptivity toward others (communication) | 0.823 | 0.193 | 0.114 | 0.230 | 0.022 |
| Relational adjustment (communication) | 0.786 | 0.185 | 0.135 | 0.209 | 0.109 |
| Kindness | 0.762 | 0.240 | 0.143 | 0.203 | −0.004 |
| Self-advocacy (communication) | 0.756 | 0.089 | 0.140 | 0.149 | 0.233 |
| Self-control (communication) | 0.700 | 0.199 | 0.170 | 0.344 | 0.114 |
| Thoughtfulness | 0.683 | 0.179 | 0.312 | 0.225 | 0.178 |
| Expressiveness (communication) | 0.665 | 0.140 | 0.287 | 0.092 | 0.357 |
| Sense of humor | 0.642 | 0.302 | 0.240 | −0.072 | 0.214 |
| Stress tolerance | 0.550 | 0.082 | 0.135 | 0.533 | 0.085 |
| Wit | 0.488 | 0.136 | 0.295 | 0.162 | 0.478 |
| Deduction | 0.472 | 0.099 | 0.457 | 0.224 | 0.296 |
| Muscular strength | 0.018 | 0.885 | 0.041 | 0.036 | 0.053 |
| Agility | 0.181 | 0.723 | 0.099 | 0.147 | 0.291 |
| Balance | 0.326 | 0.669 | 0.087 | 0.201 | 0.120 |
| Flexibility | 0.199 | 0.623 | 0.121 | 0.123 | 0.240 |
| Attractiveness/proportionality | 0.354 | 0.583 | 0.369 | −0.150 | 0.050 |
| Endurance | 0.142 | 0.546 | 0.173 | 0.464 | 0.077 |
| Memory | 0.148 | 0.074 | 0.789 | 0.262 | 0.104 |
| Calculation ability | 0.186 | 0.225 | 0.699 | 0.193 | 0.061 |
| Comprehension | 0.347 | −0.013 | 0.617 | 0.325 | 0.322 |
| Foreign language | 0.125 | 0.207 | 0.609 | −0.144 | 0.059 |
| Reading comprehension (communication) | 0.527 | 0.001 | 0.568 | 0.251 | 0.216 |
| Patience | 0.464 | 0.163 | 0.072 | 0.722 | 0.075 |
| Concentration | 0.114 | 0.172 | 0.377 | 0.567 | 0.307 |
| Aspiration | 0.247 | 0.091 | 0.166 | 0.515 | 0.358 |
| Belief | 0.394 | 0.143 | 0.082 | 0.475 | 0.248 |
| Performance | 0.050 | 0.259 | 0.054 | 0.166 | 0.687 |
| Sensitivity | 0.305 | 0.220 | 0.214 | 0.191 | 0.587 |
| Skill level | 0.261 | 0.419 | 0.255 | 0.137 | 0.463 |

Iterative principal factor analysis with polychoric correlation coefficients and varimax rotation; *n* = 804.

### 3.5. Linear Regression Analysis (Table 5)

Each of the five FSLs was entered as an objective variable in the linear regression model with the distinction between LE and SE, the setting of the experience, sex, age, highest academic degree, annual household income, and residential area set as explanatory variables (Table 5). The variable most strongly associated with the various FSLs was the setting of the experience. For work/housework/childrearing, relational FSL was high, whereas sports activities were related to high physical and low cognitive FSLs. Arts and cultural activities showed low psychological and high technical skills FSL. In addition, both relational and physical FSLs were lower for SE than for LE. After adjusting for these variables—which were related to the experiences—differences in FSL were examined as associated with individual characteristics. Female respondents had lower cognitive and higher psychological FSLs than male respondents. Additionally, psychological FSL was lower in the older age group than in the younger age group. Those with higher academic degrees had lower physical and cognitive FSL than those with lower academic degrees, and a low relational FSL was noted for the middle-income tier. Residential location (urban versus non-urban) showed no significant differences for any FSL.

**Table 5.** Respondent characteristics as associated with their experience with limits and FSL.

| | | Relational FSL | | Physical FSL | | Cognitive FSL | | Psychological FSL | | Technical Skills FSL | |
|---|---|---|---|---|---|---|---|---|---|---|---|
| | | Coefficient | *p*-Value | Coefficient | *p*-Value | Coefficient | *p*-Value | Coefficient | *p*-Value | Coefficient | *p*-Value |
| Experience | LE (reference) | | | | | | | | | | |
| | SE | −0.158 * | 0.022 | −0.150 * | 0.019 | 0.018 | 0.775 | −0.054 | 0.435 | −0.058 | 0.377 |
| Experience setting | Work/housework/ childrearing (reference) | | | | | | | | | | |
| | Academics | −0.528 ** | <0.001 | −0.049 | 0.548 | 0.719 ** | <0.001 | −0.057 | 0.521 | 0.124 | 0.138 |
| | Sports activities | −0.738 ** | <0.001 | 0.942 ** | <0.001 | −0.289 ** | 0.001 | −0.126 | 0.186 | 0.220 * | 0.015 |
| | Arts and cultural activities | −0.387 ** | 0.005 | 0.003 | 0.983 | −0.066 | 0.599 | −0.483 ** | 0.001 | 0.782 ** | <0.001 |
| | Other | −0.020 | 0.884 | 0.195 | 0.118 | −0.049 | 0.694 | −0.314 * | 0.021 | 0.178 | 0.166 |
| Sex | Male (reference) | | | | | | | | | | |
| | Female | 0.017 | 0.818 | −0.002 | 0.974 | −0.269 ** | <0.001 | 0.159 * | 0.032 | −0.115 | 0.101 |
| Age | 20–39 years (reference) | | | | | | | | | | |
| | 40–64 years | −0.053 | 0.500 | 0.095 | 0.200 | 0.086 | 0.238 | −0.086 | 0.282 | −0.104 | 0.168 |
| | ≥65 years | −0.118 | 0.212 | 0.110 | 0.211 | −0.010 | 0.909 | −0.226 * | 0.019 | −0.042 | 0.641 |
| Highest academic degree | Junior high/high school (reference) | | | | | | | | | | |
| | Trade school, 2-year college, or higher professional school | −0.147 | 0.139 | −0.011 | 0.906 | −0.112 | 0.223 | 0.151 | 0.133 | 0.055 | 0.560 |
| | University/graduate school | −0.154 | 0.064 | −0.167 * | 0.030 | −0.203 ** | 0.008 | −0.028 | 0.735 | 0.085 | 0.286 |
| Household annual income | <JPY 4 million (reference) | | | | | | | | | | |
| | JPY 4–8 million | −0.184 * | 0.019 | −0.098 | 0.177 | −0.008 | 0.910 | 0.003 | 0.972 | −0.038 | 0.610 |
| | ≥JPY 8 million | −0.152 | 0.110 | −0.075 | 0.400 | 0.019 | 0.834 | 0.076 | 0.429 | −0.031 | 0.737 |
| Residential area | Non-urban setting (reference) | | | | | | | | | | |
| | Urban setting | −0.095 | 0.185 | −0.002 | 0.976 | 0.112 | 0.091 | −0.030 | 0.676 | −0.051 | 0.454 |

Linear regression model with each factor score set as an objective variable. $n$ = 648. *: $p < 0.05$, **: $p < 0.01$

## 4. Discussion

In this study, we aimed to (1) describe individual experiences with the presence or absence of sensing limits, (2) investigate the different ways in which limits can be sensed (factors of the sense of limits: "FSLs"), and (3) identify factors that correlate with the strength of FSLs. As a result, five FSLs were extracted, and variables were assessed for their associations with the strength of each FSL. The highest regression coefficients were found for physical FSL and sports activities (0.942), cognitive FSL and academics (0.719), psychological FSL and arts and cultural activities (−0.483), and technical skills FSL and arts and cultural activities (0.782). Meanwhile, relational FSL had low regression coefficients with academics (−0.528), sports activities (−0.738), and arts and cultural activities (−0.387), suggesting that for these activities, respondents did not feel that they had reached a limit in their communication abilities, stress tolerance, kindness, thoughtfulness, or sense of humor.

When examined by sex, women were significantly less limited with respect to cognitive FSL (−0.269), which deals with memory, calculation ability, comprehension, foreign languages, and reading comprehension (communication), whereas they felt more limited with psychological FSL (0.159), which deals with patience, concentration, aspiration, and belief, relative to men. This sex-dependent difference may be indicative of the critical issue of women's status within societies wherein women are often convinced that they lack skills. Although the underlying causes of this in the Japanese societal structure are certainly important, they are beyond the scope of this study. Accordingly, the discussion below focuses solely on the data obtained within the present study, without extrapolation.

With regard to age, older individuals had significantly lower psychological FSL (−0.226), indicating that they felt less limited in their capacity for patience, concentration, aspiration, and beliefs. Meanwhile, those with higher academic degrees had significantly lower cognitive (−0.203; memory, calculation ability, comprehension, foreign language) and psychological (−0.167; agility, flexibility, endurance) FSLs, demonstrating a relative lack of limits. With regard to annual income, relational FSL (−0.184) was significant for the income group of JPY 4–8 million; however, as shown in Table 2, the JPY 8 million or higher income group comprised significantly higher numbers of NE individuals (neither LE nor SE). For those who had experienced either LE or SE, the low sense of limits among the middle-income tier may have partly been because of stress tolerance and receptivity toward others, among other items encompassed by relational FSL.

The results in Section 3.3 are noteworthy. Respondents who answered "Only try to do what can be done at that given moment," had already developed a method to cope with stress. An interview survey of these respondents may shed light on how others might learn to develop their own coping strategies. Meanwhile, there were no age-dependent trends among respondents who answered, "Never had any goal or objective for which there was a desire to fulfill to the point of striving to the maximum limit" (data not shown), which may symbolize the apathetic strata in current Japanese society. Opinions are likely to be divided based on whether intervention is necessary for this group. However, when formulating social policies, it will be important to understand the various FSLs in each country. For example, a tax levy system that would push people past their limits would not succeed as a national strategy.

### 4.1. Philosophical Implications of LE or SE

There are two possible philosophical implications of the description of LE or SE for the neuroethics of enhancement.

First, it allows for a more detailed discussion of the motivations of enhancement users. As shown in Section 3.5, the fact that we were able to extract five explanatory factors, namely relational, physical, cognitive, psychological, and technical skills FSLs, is significant in terms of philosophical implications. In previous studies on the ethics of enhancement, the motive for enhancement has been assumed to be a given; thus, it has not been the subject of discussion. In this context, structuring the motive for the act of enhancement by describing the LE or SE leads to a higher-resolution study of the ethics of enhancement. In other

words, by focusing on the motives rather than the consequences of the act, it is possible to not only discuss enhancement in a consequentialist way of thinking but also conduct a deontological approach to the ethics of enhancement. If this deontological approach can lead to an enhancement concept that allows for various types of self-realization based on the diversity of individual motives, the debate on the nature of enhancement may be renewed. Of course, this would require further normative consideration beyond the scope of this study.

Second, the extraction of the SE opens up new types of enhancement directions. The result that 32.0% of the study respondents had suppression experiences without limit experiences may have had some impact in terms of opening up the possibilities of enhancement technologies. This study shows that people try to achieve self-realization toward a goal but give up on motivation in the process. It will be possible to apply neuroenhancement technology to this structure. If we can manipulate the inhibitory function that causes us to stop striving before reaching our limits, we may be able to develop ourselves more efficiently. However, removing such inhibition may increase the potential for injury and overwork. Considering that the release of the inhibitory function of the mind and overwork are points that have not been included in previous discussions on the ethics of enhancement, the results of this study on extracting the suppression experience might be an expansion of the neuroethics of enhancement.

### 4.2. Limitations

This study had some limitations. First, the authors selected 29 items after consulting the literature and expert opinion, but the results might have been different if other items had been included. Second, as this was an online survey of respondents registered as monitors with an Internet survey company, the respondents were predominantly those with high Internet literacy, which may have created some bias.

### 5. Conclusions

To clarify the concept of limit, which is cross-applicable to the discussion of enhancement motivation, this study conducted an exploratory investigation of LE and SE. Overall, 32% of respondents had LE and another 32% had no LE but SE. Five FSLs were extracted for exploratory analysis (relational, physical, cognitive, psychological, and technical skills). LE and SE in the five FSLs were directly related to enhancement motivation. Factors correlated with FSL intensity were lower in the middle-income tier for relational FSL, lower in the more educated tier for physical FSL, lower in female respondents and the more educated tier for cognitive FSL, and lower in male respondents and the older age groups for psychological FSL. Thus, the five FSLs were perceived differently by varied attributes. Furthermore, 36% of the total respondents did not have LE or SE. The findings also suggest that respondents' experience structure can be explained in terms of coping with stress. Further detailed studies, including qualitative research on the process leading to LE and SE, are required. As exploratory research, this study takes a step toward elucidating enhancement motivation, and it may have the potential to influence future trends in the development of enhancement technologies. We must enhance our understanding of this concept in a philosophically robust manner.

**Author Contributions:** Conceptualization, E.N. and A.A.; methodology, E.N.; software, K.M.; validation, E.N. and K.M.; formal analysis, K.M.; investigation, E.N.; data curation, E.N.; writing—original draft preparation, A.A.; writing—review and editing; E.N., K.M. and A.A.; supervision, A.A.; project administration, A.A.; funding acquisition, E.N. All authors have read and agreed to the published version of the manuscript.

**Funding:** This study was supported by a grant from the Japan Society for the Promotion of Science (JSPS KAKENHI JP21K21106).

**Institutional Review Board Statement:** This study received approval from the University of Tokyo Research Ethics Committee (review No. 10950).

**Informed Consent Statement:** Written informed consent is not necessary for this type of Internet study.

**Data Availability Statement:** Not applicable.

**Conflicts of Interest:** The authors declare no conflict of interest.

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
