# Peer review of "Individual Experiences with Being Pushed to Limits and Variables That Influence the Strength to Which These Are Felt: A Cross-Sectional Survey Study"

_2571-8800, doi:10.3390/j5030024_

Round 1

Reviewer 1 Report

Dear authors, thank you for the opportunity to get acquainted with your work.

Below are detailed comments reflecting the need for revision of your manuscript:

1. Introduction. Fragmentary, does not reflect a detailed analysis of the problem, needs to be improved. The goal setting needs to be improved. It is written as a series of steps and tasks. There are no hypotheses and research questions and their justification.

2. It is required to reflect in detail the structure of the questionnaire, what number of questions for each block (attach the questionnaire itself to the article). Indicate the rationale for choosing each block of questions. Has the reliability and validity of the instrument been checked?

3. The discussion of the results contains insufficient interpretation of the obtained data, as well as their correlation with the results of other studies. Not conceptualized enough. There are no further steps in the development of this study.

4. Conclusions are very scarce, do not reflect all the results obtained. Requires expansion and detail.

5. The list of references needs to be expanded. Represented by only 20 sources. Of these, only 2 in the last 5 years. Which allows us to conclude that the issue has not been adequately studied.

In connection with the above, unfortunately, the article cannot be recommended for publication.

best regards, the reviewer

Author Response

Response to Reviewer 1

Dear authors, thank you for the opportunity to get acquainted with your work.

Below are detailed comments reflecting the need for revision of your manuscript:

Thank you for reading our manuscript. We have significantly revised the original manuscript since reference 1 regarding neuroenhancement has been published. 

Originally, this first paper is a base for the ref.1, neuroenhancement study (attached in Appendix B), so I have confused you quite a bit. I apology it. So, you can read re-submitted manuscript as a completely new article.

This is a really challenging trial to establish a new concept.

  1. Nakazawa, E.; Mori, K.; Udagawa, M.; Akabayashi, A. A Cross-Sectional Study of Attitudes towards Willingness to Use Enhancement Technologies: Implications for Technology Regulation and Ethics. BioTech 2022, 11, 21. https://doi.org/10.3390/ biotech11030021

  1. Fragmentary, does not reflect a detailed analysis of the problem, needs to be improved. The goal setting needs to be improved. It is written as a series of steps and tasks. There are no hypotheses and research questions and their justification.

We have significantly revised. Please refer to reference 1 and you will understand the background of this paper.

  1. It is required to reflect in detail the structure of the questionnaire, what number of questions for each block (attach the questionnaire itself to the article). Indicate the rationale for choosing each block of questions. Has the reliability and validity of the instrument been checked?

All are described in the new manuscript. Since this is an innovative exploring survey, there are little previous studies.

Attach the questionnaire itself to the article⇒ The survey has been done in Japanese. If you see the method section and Table 4, you will understand how the questionnaire looks like. If you still desire, we will translate it into Japanese and attach the translation, although we think it will not add more important necessarily information.

  1. The discussion of the results contains insufficient interpretation of the obtained data, as well as their correlation with the results of other studies. Not conceptualized enough. There are no further steps in the development of this study.

Please read the new manuscript with reference 1. You will understand why we need to do this study.

  1. Conclusions are very scarce, do not reflect all the results obtained. Requires expansion and detail.

As this is an exploratory data, we cannot speculate too much. Rather we emphasize the necessity of further studies.

  1. The list of references needs to be expanded. Represented by only 20 sources. Of these, only 2 in the last 5 years. Which allows us to conclude that the issue has not been adequately studied.

Since this is an innovative exploring survey, there are little previous studies. And the most recent our study in 2022 is added. Again, this is a challenging trial to establish a new concept.

In connection with the above, unfortunately, the article cannot be recommended for publication.

best regards, the reviewer

We hope you will understand the aim of the study written in new version and hope you will be comfortable with publishing our innovative paper. Once again, thank you for reading our paper in detail.

Reviewer 2 Report

Your work is interesting. I hope my comments help. I believe more explanations are required.

Abstract, I believe the MDPI format asks to not use Background: Methods: etc. It should just be a continuous paragraph.

Keywords, I suggest some of your important variables – physical FSL, sport, academics, medical care settings or environments

Power harassment needs a definition on page 1.

Introduction, it seems short. Concise is a good way to write. It seems you should expand upon references 6-10 in a paragraph or two.

Methods, should it be to complete it was (not were)

Methods, I do not know what a monitor is.

Methods, I think a supplement file with the questionnaire is needed. Or even a table of the possible options.

2.2. Analysis, you need more details for why you chose the specific analyses.

Table 1, I do not understand how you could sample as you wrote “in a manner that would yield roughly…”

What is roughly?

How did you get the experience with limits to be so similar?

At times you have P and then you have p-value.

Discussion, more discussion please. It reads as an extended results section.

Clinical applications, I do not know from where you pulled this application from your data. There needs to be more in the introduction to get the reader to the point of your application. It seems to come out of nowhere.

Author Response

Response to Reviewer 2

Comments and Suggestions for Authors

Thank you for reading our manuscript. We have significantly revised the original manuscript since reference 1 regarding neuroenhancement has been published. 

Originally, this first paper is a base for the ref.1, neuroenhancement study (attached in Appendix B), so I have confused you quite a bit. I apology it. So, you can read re-submitted manuscript as a completely new article.

This is a really challenging trial to establish a new concept.

Nakazawa, E.; Mori, K.; Udagawa, M.; Akabayashi, A. A Cross-Sectional Study of Attitudes towards Willingness to Use Enhancement Technologies: Implications for Technology Regulation and Ethics. BioTech 2022, 11, 21. https://doi.org/10.3390/ biotech11030021

Your work is interesting. I hope my comments help. I believe more explanations are required.

Abstract, I believe the MDPI format asks to not use Background: Methods: etc. It should just be a continuous paragraph.

We have changed it.

Keywords, I suggest some of your important variables – physical FSL, sport, academics, medical care settings or environments

Thank you. Added.

Power harassment needs a definition on page 1.

We did not use the term in the revised version.

Introduction, it seems short. Concise is a good way to write. It seems you should expand upon references 6-10 in a paragraph or two.

Please read the revised version. If it is still short, we will add.

Methods, should it be to complete it was (not were)

It is now completed.

Methods, I do not know what a monitor is.

Explained: who were registered as questionnaire response members (monitors) of a major Internet survey company

Methods, I think a supplement file with the questionnaire is needed. Or even a table of the possible options.

The survey has been done in Japanese. If you see the method section and Table 4, you will understand how the questionnaire looks like. If you still desire, we will translate it into Japanese and attach the translation, although we think it will not add more important necessarily information.

2.2. Analysis, you need more details for why you chose the specific analyses.

Pearson’s χ2 test, Factor analysis, and Linear Regression Analysis are standard step to explore associate factors.

Table 1, I do not understand how you could sample as you wrote “in a manner that would yield roughly…”

We want nearly the same number of respondents in both sexes and age groups. Since this is an exploratory internet survey, we do not have hypothesis to be tested and did not calculate power. We continued recruiting until we get enough number pf respondents for statistical analysys. Accordingly, there is no response rate.

What is roughly?

Nealy the same number, not exactly the same.

How did you get the experience with limits to be so similar?

Sorry, I do not get your point. The experience with limits are not at all similar.

At times you have P and then you have p-value.

Changed to be consistent.

Discussion, more discussion please. It reads as an extended results section.

Please read the new version. If you still need more, we will try, but what we can say from the result is rather limited.

Clinical applications, I do not know from where you pulled this application from your data. There needs to be more in the introduction to get the reader to the point of your application. It seems to come out of nowhere.

Revised.

We hope you will understand the aim of the study written in new version and hope you will be comfortable with publishing our innovative paper. Once again, thank you for reading our paper in detail.

Reviewer 3 Report

Thank you for the opportunity to revise the paper titled: "Individual experiences with being pushed to limits and variables that influence the strength to which these are felt: A cross-sectional survey study".

In the section Introduction, the authors should clearly explain the research problem and the objectives of the study. The authors need to articulate the ‘missing puzzle piece’ their research aims to cover more clearly and specifically. I usually recommend the logical flow for the introduction to be: 1) what is the problem and why is it important, 2) what we know, 3) what we don’t know, and finally, 4) what are we doing about it.

Authors should add research hypotheses.

The section Discussion provides a simple summary of the outcomes, that support all the hypotheses. This however raises the question, what is new about the outcome of this research? This needs to be emphasized more clearly when putting together the literature review and the conceptual framework and discussed in the discussion/results section to clarify the contribution of the paper to the body of knowledge.

Kind regards,

reviewer

Author Response

Response to Reviewer 3

Comments and Suggestions for Authors

Thank you for the opportunity to revise the paper titled: "Individual experiences with being pushed to limits and variables that influence the strength to which these are felt: A cross-sectional survey study".

Thank you for reading our manuscript. We have significantly revised the original manuscript since reference 1 regarding neuroenhancement has been published. 

Originally, this first paper is a base for the ref.1, neuroenhancement study (attached in Appendix B), so I have confused you quite a bit. I apology it. So, you are able to read re-submitted manuscript a completely new article.

Nakazawa, E.; Mori, K.; Udagawa, M.; Akabayashi, A. A Cross-Sectional Study of Attitudes towards Willingness to Use Enhancement Technologies: Implications for Technology Regulation and Ethics. BioTech 2022, 11, 21. https://doi.org/10.3390/ biotech11030021

In the section Introduction, the authors should clearly explain the research problem and the objectives of the study. The authors need to articulate the ‘missing puzzle piece’ their research aims to cover more clearly and specifically. I usually recommend the logical flow for the introduction to be: 1) what is the problem and why is it important, 2) what we know, 3) what we don’t know, and finally, 4) what are we doing about it.

We hope you will understand when you read the new version and reference 1 attached as Appendix B. Briefly,

1) what is the problem and why is it important: For the practical use of new technology such as neuroenhancement, we have no foundation and concept for academic and public discussion and policy making.

2) what we know: We know nothing.

3) what we don’t know: We know that we do not know nothing.

4) what are we doing about it.: Conduct an exploratory study to get the bases for the establishment of the concept.

Authors should add research hypotheses.

As this is an exploratory study, we have no hypothesis to test.

The section Discussion provides a simple summary of the outcomes, that support all the hypotheses. This however raises the question, what is new about the outcome of this research? This needs to be emphasized more clearly when putting together the literature review and the conceptual framework and discussed in the discussion/results section to clarify the contribution of the paper to the body of knowledge.

Please read the new version as well as reference 1, attached as Appendix B. You will know the significance of the study.

Kind regards,

We hope you will understand the aim of the study written in new version and hope you will be comfortable with publishing our innovative paper. Once again, thank you for reading our paper in detail.

Round 2

Reviewer 1 Report

Dear authors, thank you for the significant revision of your manuscript. At the same time, a number of comments remained uncorrected in full:

1. The list of references is not sufficiently expanded (if there are few narrow studies, then it should be expanded with such works where similar problems have been investigated, whose results may be useful in analyzing your results).

2. A corollary of the first remark - the second - the discussion of the results with the works of other authors is not extended. The authors point out the novelty of the problem and the lack of similar studies. But as I noted earlier, we can see the explanation for some of the associations found in the study in other studies revealing the mechanisms. You should look for them and add them to the discussion of the results.

3. Although the authors claim that their conclusions are the most basic and they do not want to include unverified data in the conclusions, this statement is debatable. Because the purpose of the study stated by the authors is broader than the conclusion described in the conclusion (please modify the conclusion to suit the purpose and hypothesis).

In this connection, it is necessary to finalize the article before its publication.

Best regards, the reviewer

Author Response

To Reviewer 1

  1. The list of references is not sufficiently expanded (if there are few narrow studies, then it should be expanded with such works where similar problems have been investigated, whose results may be useful in analyzing your results).

Thank you for your valuable comments. We have reviewed the relevant studies again and added 24 references to the introduction.

  1. Academy of Medical Sciences, British Academy, Royal Academy of Engineering, and Royal Society. Human Enhancement and the Future of Report from a Joint Workshop Hosted by the Academy of Medical Sciences, the British Academy, the Royal Academy of Engineering and the Royal Society. The Academy of Medical Sciences: London, UK, 2012. Available online: https://acmedsci.ac.uk/viewFile/publicationDownloads/135228646747.pdf (accessed on 5 August 2022).
  2. President’s Council on Bioethics. Beyond therapy: Biotechnology and the pursuit of happiness. Available online: https://bioethicsarchive.georgetown.edu/pcbe/reports/beyondtherapy/ (accessed on 5 August 2022).
  3. Deutsches Referenzzentrum f€ur Ethik in den Biowissenschaften. 2002. Enhancement: die ethische Diskussion €uber biomedizinische Verbesserungen des Menschen. drez-Sachstandsbericht, Nr. 1. Bonn, Germany (in German).
  4. Nakazawa, E.; Yamamoto, K.; Tachibana, K.; Toda, S.; Takimoto, Y.; Akabayashi, Ethics of decoded neurofeedback in clinical research, treatment, and moral enhancement. Am J Bioeth Neurosci 2016, 7(2), 110–117. doi:10.1080/21507740.2016.1172134.
  5. Dumbili, E.W.; Gardner, J.; Degge, H.M.; Hanewinkel, R. Enhancement motivations for using prescription drugs among young adults in Nigeria. Int J Drug Policy 2021, 95, 102995. doi:1016/j.drugpo.2020.102995.
  6. Kecojevic, A.; Corliss, H.L.; Lankenau, S.E. Motivations for prescription drug misuse among young men who have sex with men (YMSM) in Philadelphia. Int J Drug Policy2015, 26(8), 764–771. doi:1016/j.drugpo.2015.03.010.
  7. Drazdowski, T.K.; Kelly, L.M.; Kliewer, W.L. Motivations for the nonmedical use of prescription drugs in a longitudinal national sample of young adults. J Subst Abuse Treat 2020,114, 108013. doi:1016/j.jsat.2020.108013.
  8. Jaspers, Way to Wisdom: An Introduction to Philosophy; Manheim, R., Trans. (1964); Yale University Press, New Haven and London, 1951; pp. 22–23.
  9. Mundt, Jaspers concept of “limit situation”: extensions and therapeutic applications. In Karl Jaspers’ Philosophy and Psychopathology; Fuchs, T., Breyer, T., Mundt, C., Eds.; Springer: New York, USA, 2014; pp. 169–178.
  10. Ikai, M.; Steinhaus, A.H. Some factors modifying the expression of human strength. J Appl Physiol 1961, 16, 157–163. doi:10.1152/jappl.1961.16.1.157.
  11. Ikai, M.; Ishi, An electromyographic study on physiological and psychological limits of human strength. Japan Journal of Physical Education, Health and Sport Sciences 1961, 5(4), 154–165. [in Japanese]
  12. Ästband, I.; Hedman, Muscular strength and aerobic capacity in men 50–64 years old. Internationale Zeitschrift für angewandte Physiologie einschließlich Arbeitsphysiologie 1963, 19, 425–429. doi:10.1007/BF00697761.
  13. Howell, M.L.; Alderman, R.B. Psychological determinants of fitness. Can Med Assoc J 1967, 96(12), 721–
  14. Ikai, M.; Yabe, Training effect of muscular endurance by means by voluntary and electrical stimulation. Internationale Zeitschrift für angewandte Physiologie einschließlich Arbeitsphysiologie 1969, 28(1), 55–60. doi:10.1007/BF00696039.
  15. Knapik, J.; Daniels, W.; Murphy, M.; Fitzgerald, P.; Drews, F.; Vogel, Physiological factors in infantry operations. Eur J Appl Physiol Occup Physiol 1990, 60(3), 233–238. doi:10.1007/BF00839165.
  16. Singh, R.; Hwa, O.C.; Roy, J.; Jin, C.W.; Ismail, S.M.; Lan, M.F.; Hiong, L.L.; Aziz, A.-R. Subjective perception of sports performance, training, sleep and dietary patterns of Malaysian junior Muslim athletes during Ramadan intermittent fasting.Asian J Sports Med 2011, 2(3), 167– doi:10.5812/asjsm.34750.
  17. Rube, N.; Secher, N.H. Paradoxical influence of encouragement on muscle fatigue. Eur J Appl Physiol Occup Physiol 1981, 46(1), 1– doi:10.1007/BF00422169.
  18. Gandevia, S.C.; Allen, G.M.; Butler, J.E.; Taylor, J.L. Supraspinal factors in human muscle fatigue: evidence for suboptimal output from the motor cortex. J Physiol 1996, 490(2), 529– doi:10.1113/jphysiol.1996.sp021164.
  19. Evans, D.R.; Boggero, I.A.; Segerstrom, S.C. The nature of self-regulatory fatigue and “ego depletion”: lessons from physical fatigue.Pers Soc Psychol Rev 2016, 20(4), 291– doi:10.1177/1088868315597841.
  20. McNair, P.J.; Depledge, J.; Brettkelly, M.; Stanley, S.N. Verbal encouragement: effects on maximum effort voluntary muscle action. Br J Sports Med 1996, 30(3), 243-245. doi:10.1136/bjsm.30.3.243.
  21. Amagliani, R.M.; Peterella, J.K.; Jung, A.P. Type of encouragement influences peak muscle force in college-age women.Int J Exerc Sci 2010, 3(4), 165–
  22. Ekblom, M.M.; Eriksson, Concurrent EMG feedback acutely improves strength and muscle activation. Eur J Appl Physiol 2012, 112(5), 1899–1905. doi:10.1007/s00421-011-2162-2.
  23. Fernandez-Del-Olmo, M.; Río-Rodríguez, D.; Iglesias-Soler, E.; Acero, R.M. Startle auditory stimuli enhance the performance of fast dynamic contractions. PLoS One 2014, 9(1), doi:10.1371/journal.pone.0087805.
  24. Takarada, Y.; Nozaki, Maximal voluntary force strengthened by the enhancement of motor system state through barely visible priming words with reward. PLoS One 2014, 9(10), e109422. doi:10.1371/journal.pone.0109422.

  1. A corollary of the first remark - the second - the discussion of the results with the works of other authors is not extended. The authors point out the novelty of the problem and the lack of similar studies. But as I noted earlier, we can see the explanation for some of the associations found in the study in other studies revealing the mechanisms. You should look for them and add them to the discussion of the results.

Thank you very much for your suggestions. It is true that there are a few existing studies. We have attempted to situate our study within the context of a few prior studies based on the points made by Reviewer 1. Since there are several studies on enhancement motivation, mainly on drug misuse, we tried to place our problem concerning “the experience of limits leading to enhancement” in that context. In addition, we tried to place the concept of limits in a historical research context. Classically, the concept of limits can be seen in the philosophical and psychological explorations of Jaspers. The concept of psychological and physiological limits can also be seen in a sports science context in a study by Ikai et al. Some of the research that continues these legacies is coming into proximity to the area of enhancement. In an effort to place our research in the history of research about limits, we have made the following additions to the introduction:

Human enhancement refers to attempts to overcome the current limits of the human body using techniques designed or applied to restore or improve human capabilities [1,2]. Human enhancement is often categorized into physical, cognitive, and moral (emotional) enhancement [3,4]. Users’ enhancement motivation is significant for the direction of enhancement technologies development. Prior studies have re-ported on enhancement motivation for cognitive and mood enhancement using drugs, arguing for the prevention of drug misuse [5–7]. However, research on enhancement motivation is still limited; thus, a comprehensive discussion is needed to extend the research to include physical enhancement and interventions using neuroscience techniques that go beyond pharmacological interventions.

 [……]

The concept of a “limit” in the definition of human enhancement can be applied to the discussion of enhancement motivation. A limit, in the broadest sense, is a point at which human effort reaches an insurmountable barrier. Individuals experience various limits in their lives.

[……]

Historically, Karl Jaspers made the concept of limit an explicit theme and examined it philosophically and psychologically. Jaspers described that “the ultimate situations—eath, chance, guilt, and the uncertainty of the world—confront me with the reality of failure. [……] And yet the Stoics’ striving is toward true philosophy. Their thought, because its source is in ultimate situations, expresses the basic drive find a revelation of true being in human failure [19].” Further, Jaspers identified the concept of limit as the fundamental energy that motives the essential life of human beings. Jaspers’ concept of limit has been reevaluated in recent years, and its psychotherapeutic implication and applicability have been indicated in some researches [20].

The concept of limit (or limitation) was also developed in the field of sports sci-ence as a content-correlated but possibly separate source. According to the classic studies of Ikai et al. [21,22], the physiological limit of muscle strength is the upper limit of muscle strength determined by structural factors, and the psychological limit is the upper limit of muscle strength determined by functional factors. Ikai et al.’s studies opened up a research area on the subjective aspect of muscle strengthening and fitness [23–27], which has resulted in research on fatigue [28–30], physical exercise, and encouragement [28,31,32]. In recent years, studies on intervening in the psychological aspects of physical exercise to improve performance, including neurofeedback interventions, have also been conducted [22–35], and Ikai et al.’s concept of limit [21,22] is becoming more relevant to contemporary enhancement technologies.

We thus assumed that the concept of limit or suppression experiences could be a useful cross-cutting concept in the discussion of motivation as a foundational theory of enhancement and attempted to clarify it.

  1. Although the authors claim that their conclusions are the most basic and they do not want to include unverified data in the conclusions, this statement is debatable. Because the purpose of the study stated by the authors is broader than the conclusion described in the conclusion (please modify the conclusion to suit the purpose and hypothesis).

Thank you. In light of your comments, we have added to and revised our conclusions to meet the objectives of this study as follows:

To clarify the concept of limit, which is cross-applicable to the discussion of enhancement motivation, this study conducted an exploratory investigation of LE and SE. Overall, 32% of respondents had LE and another 32% had no LE but SE. Five FSLs were extracted for exploratory analysis (Relational, Physical, Cognitive, Psychological, and Technical skills). LE and SE in the five FSLs were directly related to enhancement motivation. Factors correlated with FSL intensity were lower in the middle-income tier for relational FSL, lower in the more educated tier for physical FSL, lower in female respondents and the more educated tier for cognitive FSL, and lower in male respondents and the older age groups for psychological FSL. Thus, the five FSLs were perceived differently by varied attributes. Furthermore, 36% of the total respondents did not have LE or SE. The findings also suggest that respondents’ experience structure can be explained in terms of coping with stress. Further detailed studies, including qualitative research on the process leading to LE and SE, are required. As an exploratory research, this study takes a step toward elucidating enhancement motivation, and it may have the potential to influence future trends in the development of enhancement technologies. We must enhance our understanding of this concept in a philosophically robust manner.

Reviewer 2 Report

Thank you for your revised manuscript as your revisions helped me understand your first submission. I feel you replied in full.

I do see small edits that I assume the journal editors will catch. It is always hard to be perfect.

Author Response

Thank you for your revised manuscript as your revisions helped me understand your first submission. I feel you replied in full.

I do see small edits that I assume the journal editors will catch. It is always hard to be perfect.

Thank you. We are sincerely grateful of your appreciation of our manuscript.
